# Explanation for Trajectory Planning using Multi-modal Large Language Model for Autonomous Driving

## Abstract

*In automatic driving, it is important to convey the intention of the driving behavior of the ego car to the driver or passengers to achieve reliable and trustworthy driving. Most conventional methods that explain the intention and justification of the ego car use only image input without trajectory information, which is insufficient for explaining the intention of the ego car. In this study, we propose a multi-modal large language model based explanation method for trajectory planning that uses not only the frontal image but also the trajectory planning information of the ego car as input. Based on a dedicated dataset in which both the frontal video and trajectory planning information are simultaneously acquired, we confirm that this method can give effective results compared with the case without trajectory information.*

## 1. Introduction

In autonomous driving, it is important to increase the interpretability of the vehicle behavior to realize trustworthy autonomous driving for the driver. Particularly, the explanation of the behavior using language deepens human understanding. It not only enhances the driver's trust and social acceptance from a social perspective, but also contributes to deepening the system understanding of developers and researchers from the perspective of the development process [2].

BDD-X [5] is a pioneering study that describes and justifies driving behavior. BDD-X builds a driving behavior caption dataset comprising approximately 40 s of video, acceleration and direction information, and approximately 7,000 textual descriptions of the behavior and justifications for the behavior.

ADAPT [4] is a study utilizing the BDD-X driving behavior caption dataset. ADAPT simultaneously optimizes the control signal prediction and driving caption generation of the vehicle. From the viewpoint of generating a driving action caption, the input is only a recorded video without any control signal. However, in practice, the input information was insufficient to generate descriptions and justifications for actions from video information only.

In recent years, studies have used not only video information but also control information as input to generate descriptions and justifications of actions. For example, DriveGPT4 [14] uses a multi-modal large language model (LLM), and the inputs of the multi-modal LLM are a video token, the text of a question, a past control value, the text of an answer, and a next control value. However, because BDD-X is a dataset cpmprising video and control information (acceleration and direction), and the description/justification of behavior at each instance, it is a suitable model for explaining the description/justification of behavior at the current time based on a past control value sequence. Although it is possible to generate captions for explanations and justifications for past actions, it is not possible to generate captions for explanations and justifications of future trajectories (intentions).

Therefore, herein, we aim to generate captions for explaining the future trajectory (intention). If the trajectory planning information can be explained to the driver, the sense of security and trustworthiness for autonomous driving can be enhanced. However, because there is no dataset available or publicly available to obtain the trajectory planning information at each instance, it is necessary to collect the data independently. In conventional methods, past control information was input to the multi-modal LLM as text information. Herein, we propose a method, in which we train and link the video and trajectory information, thereby improving the accuracy of caption generation.

The proposed method makes three contributions:
- It generates captions for explanations and justifications of future trajectories (intentions).
- An overlaid method is built to link and train video and trajectory information.
- We compile and annotate a new dataset consisting of video and captions for trajectory planning information.

This paper first describes related studies and then de-

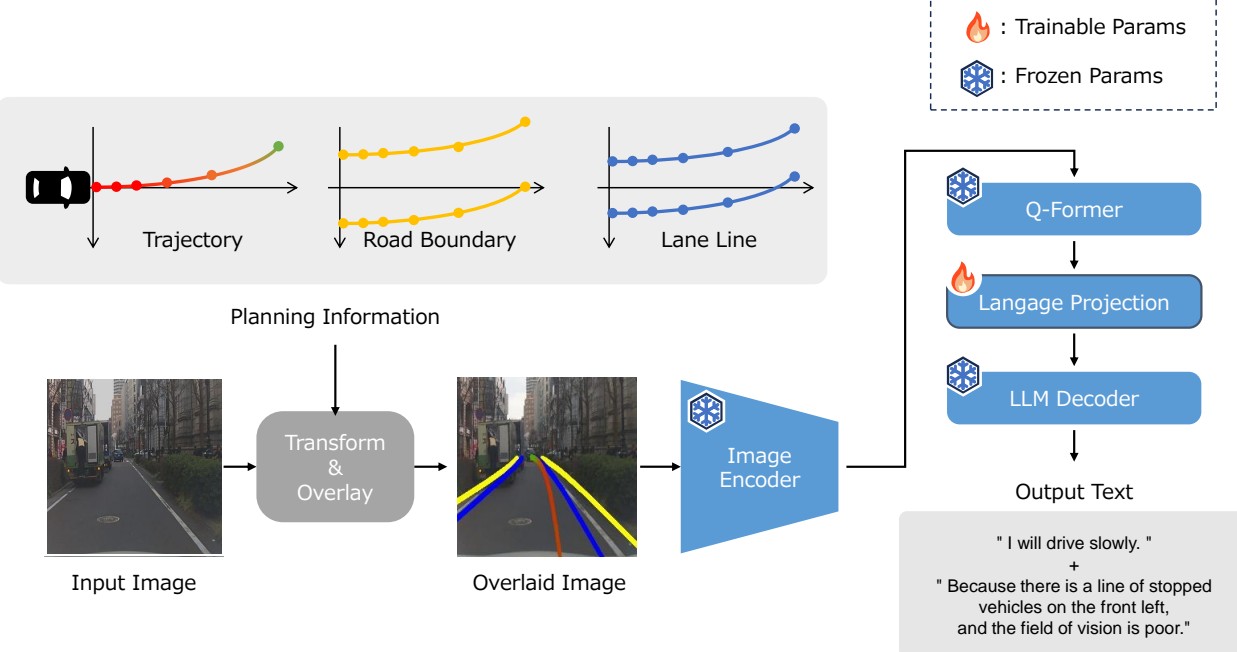

Figure 1. The pipeline of the proposed method

scribes the proposed method. The dataset, quantitative evaluation results, and qualitative evaluation results are described as the experimental results and conclusion.

## 2. Related works

This section describes the related studies of caption generation as well as a study on a driving behavior caption dataset.

### 2.1. Caption generation

Caption generation is a task to generate text describing an image by inputting the image, and Show and Tell [12] and Neural Baby Talk[7] using LSTM[3] have been proposed. In recent years, the performance of transformers has improved dramatically as datasets have become larger. BLIP2 [6], A well-known model for caption generation, leverages the existing vision encoder and LLM model, and combines them via a transformer-based network called the Q-Former to achieve image caption generation and visual question answering. The study aimed to reduce the cost of training and maintain the performance of the vision encoder and LLM by training only the Q-Former. It is a pioneering research that is the basis of several applied studies.

Moreover, a study was conducted on the generation of captions about driving behavior in ADAPT [4]. ADAPT simultaneously optimizes the control signal prediction and driving caption generation of the vehicle. It optimizes a module that uses a transformer to convert a video into a video token and uses the video token as input to predict acceleration and direction information, and a module that generates the description and justification of the action by multi-task learning.

In recent years, studies have used not only video information but also control information as input to generate descriptions and justifications of actions. For example, DriveGPT4 [14] uses a multi-modal LLM, and the inputs of the multi-modal LLM are a video token, the text of a question, a past control value, the text of an answer, and a next control value. However, because BDD-X is a dataset cpmprising video and control information (acceleration and direction), and the description/justification of behavior at each instance, it is a suitable model for explaining the description/justification of behavior at the current time based on a past control value sequence.

In this paper, we generate captions about explanations and justifications for future trajectories (intentions).

### 2.2. Driving action caption dataset

BDD-X [5] is a pioneering study that describes and justifies of driving behavior. BDD-X builds a driving behavior caption dataset consisting of approximately 40 s of video, acceleration and direction information, as well as approximately 7,000 textual descriptions of behavior and justifications for behavior. For example, the description of the driving behavior can be "The car moves back into the left lane," and the justification for that driving behavior can be "because the school bus in front of it is stopping".

In recent years, research has been conducted to describe scenes more comprehensively. For example, in DriveLM [10], the question & answering (QA) for the object, situation, and control instructions for the vehicle are described in text. It is a dataset in which the text is added to a dataset with three-dimensional annotations referred to as nuScenes [1]. In addition, nuScenesQA [8] describes the QA to an

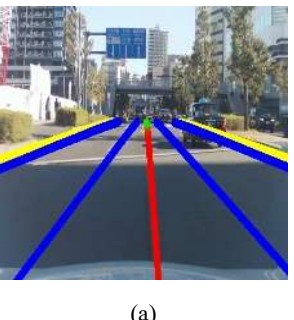 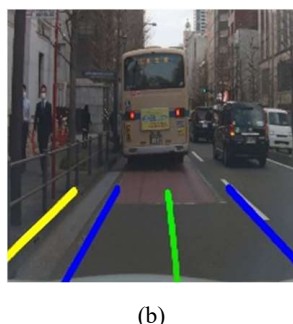 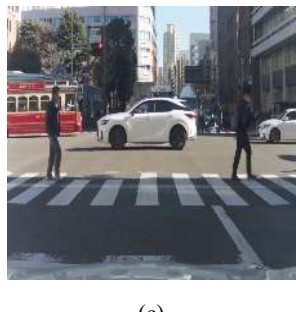 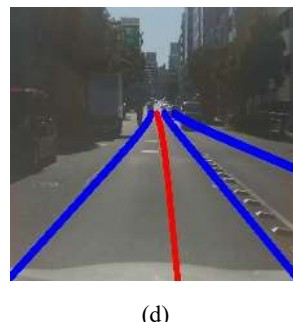

(a)         (b)         (c)         (d)

(a) I will drive at a steady speed. , Because there is a safe distance from the front vehicle.

(b) I will slow down. , Because the front vehicle is stopped.

(c) I will maintain the parked state. , Because the traffic light is red.

(d) I will drive at a steady speed. , Because there is a safe distance from the front vehicle.

Figure 2. The examples of the dedicated dataset

object. Many studies have recently been conducted to generate captions in nuScenes such as adding text to nuScenes as a base. However, these studies are only concerned with the objects and situations in a scene, and objects and control instructions to be observed. Only the BDD-X dataset explains the behavior of the vehicle and its justification.

In this study, we generate captions on explanations and justifications for future trajectories (intentions). However, because there is no dataset that can acquire the future trajectory (intention) at each instance, we collect the data independently, and establish a new dataset.

## 3. Method

### 3.1. Overview

In autonomous driving, it is important to increase the interpretability of the vehicle behavior to realize trustworthy autonomous driving for the driver. Here we describe the pipeline of our method that takes the image and trajectory planned by the autonomous driving system.

### 3.2. Pipeline

Fig. 1 shows the pipeline of the proposed method. In the proposed method, not only frontal camera image of the ego car but also trajectory planning information are used as inputs. The planning information consists of the trajectory, a pair of road boundaries and a pair of lane lines, and it is transformed and overlaid on the frontal image. Herein, we draw each line in different colors. For examples, we draw a pair of road boundary in yellow, a pair of lane line in blue. Furthermore, to keep the velocity information of the trajectory we vary the color of the trajectory line with the velocity; fast in red and slow in green. The overlaid images are encoded by image encoder. Owing to the robustness when dealing with an out-of-distribution sample[9, 11], we use

the CLIP visual encoder as an image encoder. Because the CLIP visual encoder needs to be fine-tuned carefully[13], we decided to freeze the parameters of the visual encoder during the training phase. After that, Q-Former, language projection and LLM decoder based on BLIP2 are processed, and outputs the explanation and justification of the action.

### 3.3. Trajectory overlaid images

In this subsection, we introduce the detail of trajectory overlaid images. The trajectories $P_i = ((x_1, y_1), ..., (x_N, y_N)); i = (1, ..., T)$ on the $i^{th}$ frame consist of a group spatial coordinate on the Cartesian coordinate generated from the autonomous vehicle. $T$ represents the total frame number and $N$ represents the number of coordinates on the future trajectory. In DriveGPT4, trajectory information is treated as text information to input into a multi-modal LLM, but we decided to directly convert the spatial coordinate on the ground to image coordinate by using perspective projection transform after translation and rotation operations. Then we connect the points on image coordinates and draw a line on the image.

## 4. Experiment

### 4.1. Implementation

We compared the proposal method with the baseline method. A model for generating captions only from images was used as the baseline method. A model for generating a caption from an image overlaid by trajectory planning information was used as the proposal method. The same implementation method as BLIP2 was employed for both baseline and overlaid methods except for the input image. The number of training epochs is approximately 5 for both methods. In training, the parameters of image encoder, Q-Former and LLM decoder are frozen.

## 4.2. Dataset

In this study, we generate explanation and justification captions for trajectory planning (intentions). However, because there is no dataset that can acquire the trajectory planning information at each instance, it is necessary to collect the data independently. This study constructs a new dataset of its own.

For the hardware configuration, a comma 3X device developed by Comma.ai, which can easily generate the trajectory, was installed at the front of the vehicle to verify the value of adding the trajectory planning information. The frontal image and the trajectory planning informaton were simultaneously acquired. The data collected in the urban district of Japan for approximately 120 min was used as the driving data. It consists of 69 min on main roads where vehicles mainly drive, and 51 min on narrow streets where pedestrians may cross. For each image data, the explanation and justification of the action were described in English. Fig. 2 shows the examples of our dedicated dataset.

## 4.3. Quantitative evaluation

The generated sentences were evaluated using BLEU-4 and ROUGE-L, which are general metrics of the caption generation task. Each metric aims to measure a distinct aspect of the generated sentences: BLEU-4 assesses the precision of the generated sentences, where ROUGE-L evaluates their recall. The generated sentences were evaluated for explanation of action and justification each. The results obtained are presented in Tab. 1.

In the proposed method, the results were better than the baseline method for all tasks and metrics. This is because, since the trajectory information can be used in addition to the image, action description generation can be performed in consideration of the trajectory planning information that are difficult to read directly from the image.

| Model | Action | | Justification | |
|---|---|---|---|---|
| | B-4↑ | R-L↑ | B-4↑ | R-L↑ |
| Baseline | 0.40 | 0.57 | 0.23 | 0.48 |
| Proposed | **0.44** | **0.61** | **0.26** | **0.50** |

Table 1. Evaluation results of three methods on two metrics (B-4: BLEU-4, R-L: ROUGE-L)

## 4.4. Qualitative evaluation

Fig. 3 shows the difference between the results of the baseline method and that of the proposed method. In the scene shown in the figure, the vehicle is moving straight ahead diagonally to the right, and is decelerating due to the congestion ahead. Because a single image is input to the model in both the baseline method and the proposed method, they cannot estimate that the vehicle ahead is decelerating. As a result, these methods erroneously assume that the ego vehicle is running at a constant speed.

Fig. 3 shows the differences between the results of the proposed method and the baseline method. In the scenario shown in the figure, the vehicle is moving straight ahead diagonally to the right, and is decelerating due to the congestion ahead. Because the trajectory planning information is not utilized, the baseline method misinterpreted that the ego car continues to remain stationary. On the other hand, the proposed method using trajectory planning information on the overlaid image was able to estimate that the ego car is in driving and decelerating. This implies the proposed method understands that the vehicle is decelerating because the line representing the trajectory planning gradually changes from yellow to green.

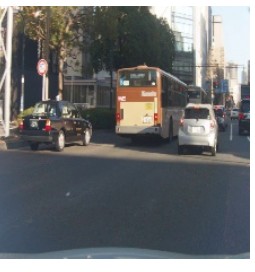 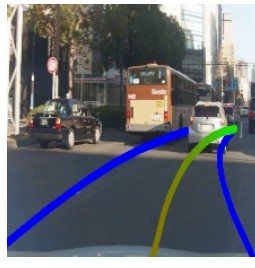

|     original image     |     overlaid image     |
|---|---|

**GT:** "I will slow down. Because the front vehicle slowed down."

**Baseline:** "I will maintain the parked state. Because the front vehicle is stopped."

**Proposed:** "I will drive slowly. Because the front vehicle is stopped."

Figure 3. Examples of generated results

# 5. Conclusion

It is important to increase the interpretability of the driving behavior in autonomous driving to realize the driver's trustworthiness. Herein, we developed a method to generate captions for explanations and justifications for trajectory planning information (intentions). Because there was no dataset that could acquire the trajectory planning information (intention) at each instance, we collected the data independently and constructed the dataset. In addition, in the conventional method, past control information was input to the multi-modal LLM as text information; However, in the proposed method, the accuracy of the caption generation of justification was improved by overlaying the video information with the trajectory planning information. Therefore, it was confirmed that the proposed method can give effective results compared with the case without trajectory information.

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
