# OpenReview forum: "Explanation for Trajectory Planning using Multi-modal Large Language Model for Autonomous Driving"
_thecvf.com/CVPR/2024/Workshop/VLADR — Submitted to VLADR 2024_

### Official Review · Reviewer_qSWu · 2024-04-21

**Rating:** 4
**Confidence:** 4

**Review:**

The work introduces a multi-modal approach that incorporates both video and trajectory planning information to generate explanations for the autonomous vehicle's future maneuvers. This method addresses the limitations of previous systems that relied solely on visual data, offering a richer, more comprehensive dataset for interpretation.

Pros:
1. This work facilitates better interpretation and explanation of driving decisions with frontal image and planned trajectory.
2. A new dataset is collected by acquiring the frontal image and trajectory planning information simultaneously.

Cons:
1. The problem formulation is questionable. I agree that the interpretability of autonomous driving systems is important, but I think a more effective way would be to optimize the language interpretation module and the trajectory planning module jointly in a chain-of-thought manner instead of facilitating interpretability with an add-on module.
2. Since the training dataset is collected only by expert human drivers with correct planning trajectories, the VLM is trained to only provide reasonable interpretation for the planned trajectory. In the case that the autonomous driving system makes a wrong decision, won’t the trained VLM still come up with an explanation to make the wrong decision look reasonable?
3. The motivation for collecting a new dataset is also unclear. There are a lot of existing large-scale driving datasets, such as nuScenes, that provide both image observations and trajectories. Language annotations can be directly achieved upon these existing datasets. Why should we collect a new dataset that is small?